# Emerging dominance of summer rainfall driving High Arctic terrestrial-aquatic connectivity

C. R. Beel [1,2,5✉], J. K. Heslop[1,3,5], J. F. Orwin [1,4], M. A. Pope[1], A. J. Schevers[1], J. K. Y. Hung [1], M. J. Lafrenière [1] & S. F. Lamoureux [1]

Hydrological transformations induced by climate warming are causing Arctic annual fluvial energy to shift from skewed (snowmelt-dominated) to multimodal (snowmelt- and rainfall-dominated) distributions. We integrated decade-long hydrometeorological and biogeochemical data from the High Arctic to show that shifts in the timing and magnitude of annual discharge patterns and stream power budgets are causing Arctic material transfer regimes to undergo fundamental changes. Increased late summer rainfall enhanced terrestrial-aquatic connectivity for dissolved and particulate material fluxes. Permafrost disturbances (<3% of the watersheds' areal extent) reduced watershed-scale dissolved organic carbon export, offsetting concurrent increased export in undisturbed watersheds. To overcome the watersheds' buffering capacity for transferring particulate material (30 ± 9 Watt), rainfall events had to increase by an order of magnitude, indicating the landscape is primed for accelerated geomorphological change when future rainfall magnitudes and consequent pluvial responses exceed the current buffering capacity of the terrestrial-aquatic continuum.

[1] Department of Geography and Planning, Queen's University, Kingston, ON, Canada. [2] Water Management and Monitoring, Environment and Natural Resources, Government of Northwest Territories, Yellowknife, NT, Canada. [3] Section 3.7 Geomicrobiology, Helmholtz Centre Potsdam GFZ German Research Centre for Geosciences, Potsdam, Germany. [4] Resource Stewardship Division, Alberta Environment and Parks, Government of Alberta, Calgary, AB, Canada. [5] These authors contributed equally: C.R. Beel, J.K. Heslop. ✉email: Casey_Beel@gov.nt.ca

Despite recent advances in our understanding of terrestrial-aquatic connectivity in permafrost-underlain watersheds[1–3], few studies directly link the controls and changes in High Arctic terrestrial-aquatic connectivity to climate warming-induced changes in hydrology[1,3–6]. As the Arctic warms and precipitation patterns change[7,8], shifts in the timing and magnitude of fluvial energy (e.g., stream power) in response to rainfall inputs will alter landscape connectivity in watersheds by increasing the late-season delivery of organic (e.g., organic carbon; OC) and inorganic (e.g., sediment, major ions) terrestrial materials into downstream aquatic ecosystems[9–11]. The impacts of shifts in the timing and magnitude of fluvial energy coupled with terrestrial ecosystem changes are scarcely documented in these environments due to a lack of integrated longer-term (≥10 years) biogeochemical records[12] and the remote nature of the High Arctic[13]. Accurate biogeochemical budgets across full hydrological seasons are critical reference points for earth system models and are required to predict the strength and timing of climate feedback mechanisms in the Arctic, including the permafrost carbon feedback[1–3,5,14]. Therefore, quantifying the impacts of the altered timing and magnitude of hydrological connectivity in these environments is necessary to parameterize earth system models that predict global change[15].

The increasing importance of the under-studied and under-represented "shoulder seasons" (early-spring and late-fall) in annual Arctic biogeochemical budgets has been recently highlighted[5], but is not quantified for most Arctic rivers. In particular, little is known about the role late-summer rainfall plays in coupling hillslopes with stream networks. Projected increases in the magnitude and frequency of summer rainfall events[7,8] are expected to elevate terrestrial-aquatic fluxes of organic and inorganic material in Arctic watersheds[5,9,11,16]. This is because the majority of rainfall events occur when suspended sediment (SS), dissolved and particulate organic carbon (DOC and POC, respectively), nutrients, and major ions are readily available for transport due to active layer depths and terrestrial biological productivity being at their annual maximums[17–19]. However, few hydrological and biogeochemical records capture the full Arctic thaw season due to the logistical constraints imposed by working in remote locations[12]. The impact of changes in temperature and precipitation on permafrost systems is further complicated by the increasing frequency of thaw-induced geomorphological disturbance (e.g., thermokarst, active layer detachments, retrogressive thaw slumps, surface subsidence), which alter Arctic landscapes, local hydrology, biogeochemistry, and terrestrial-aquatic connectivity[20–23]. Permafrost disturbances that hydro-geomorphologically couple with fluvial systems amplify the effect of climate change through disproportionate increases in the mobilization and delivery of terrestrial material into stream networks[24,25].

Here we use data from the hydrological and biogeochemical research program at the Cape Bounty Arctic Watershed Observatory (CBAWO; 2003–2019), located in the Canadian High Arctic (74°54′N, 109°35′W; Supplementary Fig. 1), to integrate changes in the timing and magnitude of fluvial energy with organic and inorganic matter quantities and transfer along the terrestrial-aquatic continuum. We used fluxes of terrestrially sourced materials (e.g., SS, POC, DOC, major ions; 2003–2017) combined with optical dissolved organic matter (DOM) indices that indicate DOM origin (2012–2017) to infer connectivity along the terrestrial-aquatic continuum. Further, we assessed the watershed-scale impacts of localized active layer detachments (ALDs; 2007–2008) on material flux in the decade following their initiation relative to concurrent Arctic greening and shifts in the timing and magnitude of fluvial energy. Through these integrated analyses, we demonstrate shifts in the timing of higher fluvial energy to later in the thaw season and reduced influence of the

snowmelt phase on material transfer. Our findings highlight the increasing importance of rainfall in driving connectivity along the High Arctic terrestrial-aquatic continuum, providing a framework to improve earth system models and demonstrating why it is increasingly important for the research community to shift our focus to capturing the late-fall shoulder season[5].

## Results and discussion

**Shifting magnitude and distribution of fluvial energy.** The primary control on Arctic hydrology is seasonal phase changes in water form and movement, characterized by a short summer flow season between spring "break-up" (2–3 weeks following daily mean air temperatures consistently >0 °C) and fall "freeze-up" (daily mean air temperatures consistently <0 °C[26]; length of summer thaw season 81 ± 11 days at the CBAWO; mean ± 1 SD unless stated otherwise). Arctic streams and rivers have distinct nival hydrological regimes (Supplementary Fig. 2a), which are characterized by one major spring snowmelt flood period (freshet) and a rapid transition to low-flow (baseflow) during the summer as the watershed snowpack is exhausted (main CBAWO rivers: 22 ± 5 days; headwater-slope streams: 17 ± 5 days). Summer baseflow is often punctuated by rainfall-generated (pluvial) discharge, with the magnitude of peak pluvial discharge largely controlled by both the timing and magnitude of rainfall and antecedent soil moisture. At the CBAWO, rainfall events are generally of low intensity with minimal hydrological significance compared to freshet, resulting in a nival-dominated (skewed distribution) hydrological regime (Fig. 1; Supplementary Fig. 2b). In absence of late-summer rainfall, baseflow wanes through fall freeze-up with little-to-no surface runoff during winter[27].

The unique seasonality of Arctic hydrology varies with latitude and is closely linked to regional weather, and in particular regional air temperature[26]. Mean regional summer (June-August; JJA) temperatures at our study site warmed ~ 2 °C between 1948 and 2019 (Mould Bay, NT; Fig. 2a). The timing and duration of the summer flow season at the CBAWO varied with summer air temperatures ($p < 0.05$). Locally, above average summer temperatures coincided with: (i) earlier first day of nival runoff and a longer summer flow season (Fig. 2c, d; $p < 0.05$); (ii) decreased watershed snow water equivalent[10]; (iii) a general reduction in maximum nival stream power from 68.0 ± 14.6 W to 42.2 ± 19.0 W; and (iv) reduced influence of the snowmelt phase on material transfer (e.g., decreased sediment flux per unit of nival runoff)[10].

The timing and magnitude of peak nival runoff at the CBAWO is strongly controlled by snow accumulation in river channels (≥4 m locally). In years rainfall had minimal hydrological significance in summer/fall ($n = 8$; Table 1; Fig. 1; Supplementary Fig. 2b), the majority of the annual available stream power was expended by the formation of channelized runoff through the snowpack, largely in isolation from the stream bed (Supplementary Fig. 3a). This runoff channel formation expended 49 ± 15% of the annual available stream power within the first 10 days of flow (Fig. 1b). On average, the entire nival period (22 ± 5 days) expended 76 ± 13% of annual available stream power (Fig. 1b). In the warmest summer of our record (2012; 4.1 ± 3.8 °C), one-third of the annual available stream power was expended in the first 6 days of flow before surface runoff made full contact with the channel bed[28]; 86% of the annual available stream power was expended during the entire nival period (29 days) the same year.

In years with proportionately more summer/fall pluvial runoff ($n = 4$), the timing of seasonal stream power shifted from a nival-dominated (skewed) to a nival- and pluvial-dominated (multimodal) hydrological regime (Fig. 1; Supplementary Fig. 2c). The three largest recorded pluvial events during the 17-year record at CBAWO redistributed hydrological energy to later in

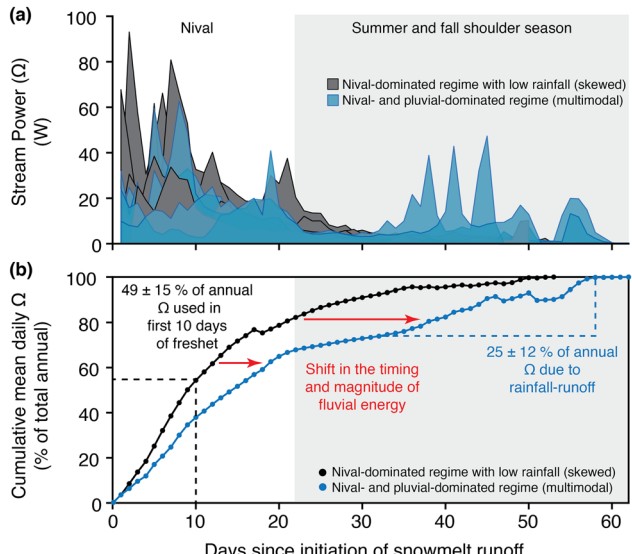

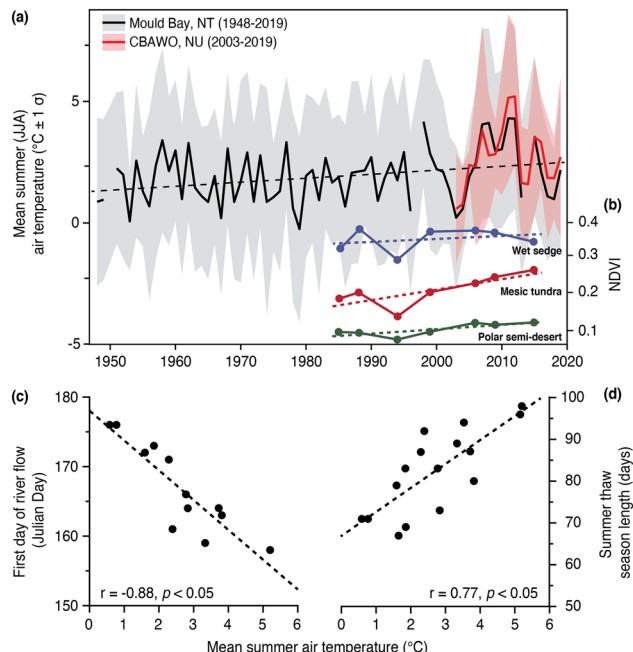

**Fig. 1 Hydrological energy budgets for small High Arctic streams. a** Daily stream power records for a small, non-glacial High Arctic stream at the CBAWO (West river; 8 km$^2$; 2003–2017) normalized to the first day of nival runoff and separated into annual hydrological regimes; (1) nival-dominated regime with little-to-no hydrological response to summer/fall rainfall inputs (skewed distribution; black/gray; $n = 8$); and (2) nival- and pluvial-dominated hydrological regime (multimodal distribution; blue; $n =$ 4). **b** Cumulative mean daily stream power (% of the total annual stream power) for the same annual hydrological regimes. In all years, 49 ± 15% of the annual available stream power is expended in the first ten days of freshet and 74 ± 13 % is expended during the full nival period (22 ± 5 days). In warmer wetter years with proportionally more pluvial discharge (blue) we observe a transition in the timing and magnitude of fluvial energy to later in the summer thaw season. In these years, pluvial runoff expended 25 ± 12% of the annual stream power budget during the summer and fall shoulder seasons. Baseflow periods had the lowest stream powers expending <5% of the annual stream power budget.

the summer (July-August) at all watershed scales (0.2–10 km$^2$) and resulted in high stream powers exceeding that of the nival peak in respective years: (i) a rain-on-snow event (late June 2007; 0.19 mm h$^{-1}$; 54 h; ≤2 year recurrence interval or >50% chance of exceedance in any given year); (ii) consecutive low-intensity rainfall (July 2007; ≤0.09–0.33 mm h$^{-1}$, <38 h, ≤2 year recurrence interval); and (iii) consecutive high-magnitude rainfall events (July 2009; 0.31–0.39 mm h$^{-1}$, 30–75 h, 8–100 year recurrence interval or <12% chance of exceedance in any given year) (Fig. 1; Supplementary Figs. 4 and 5). These pluvial responses delivered 25 ± 12% of annual available stream power to the system during only 2–5 days of runoff per event (Fig. 1b). In comparison, the entire nival period (22 ± 5 days) delivered 68 ± 15% of annual available stream power in the same years (Fig. 1b; Supplementary Fig. 4).

It is important to note that, despite the CBAWO being the longest integrative data set from the Canadian High Arctic, we have hydrological observations for only 39% of recorded rain events (Supplementary Fig. 5). This highlights the need for longer sampling seasons and/or sampling methods designed to capture late-fall shoulder season rainfall events. Nevertheless, longer-term hydrological records from circum-Arctic streams and rivers (including a northern Canadian latitudinal gradient) suggest the shift from a nival-dominated to a nival- and pluvial-dominated discharge regime is ubiquitous across both large and small watersheds of the High and Low Arctic, although the effects were dampened across larger watersheds and watersheds in the Low

**Fig. 2 Regional hydrometeorology. a** Local mean summer (Jun-Aug; JJA) air temperature for the CBAWO is significantly correlated ($r = 0.99$, $p <$ 0.05); (2003–2019) with the longer-term mean summer air temperature at Mould Bay (1948–2019; 300 km west). Warming temperatures correlated with: (**b**) increased normalized difference vegetative index (NVDI) for all vegetation types at the CBAWO (modified from Edwards and Treitz, 2017); (**c**) earlier first day of river flow and (**d**) an increase in the summer thaw season length ($p < 0.05$). Mould Bay data licensed under the Open Government License – Canada.

Arctic received proportionally more rainfall in general (8.0 to 2.95 million km$^2$ watersheds; Supplementary Figs. 6–8). In combination with our measurements from the CBAWO, these data show Arctic systems are already transitioning to a nival- and pluvial-dominated (multimodal) hydrological regime.

**Timing and magnitude of fluvial energy controls terrestrial-aquatic connectivity.** Shifts in the timing and magnitude of annual discharge patterns and stream power budgets are causing Arctic material transfer regimes to undergo fundamental changes. The term "watershed continuum" describes source-to-sink controls on material (e.g., sediment and nutrients) erosion, transfer, and deposition. The watershed continuum can be defined as a "cascading system" composed of a chain of subsystems dynamically linked by a cascade of mass and energy[29] (e.g., surface runoff from headwater-slope streams becomes part of the input into the main channel system). Internal regulators (e.g., channel slope, morphology, length) and thresholds are critical for determining if materials and energy are dissipated, stored within a subsystem, or conveyed to the adjacent subsystem. Hydrology is a key driver of the watershed continuum. Terrestrial-aquatic connectivity at the CBAWO follows a temporary storage-remobilization regime along the watershed continuum. The channel slope and morphology (alternating step-pool, riffle-pool, and braided reaches) favor energy dissipation and sediment deposition, temporarily storing materials on channel bars and the channel bed[28]. This material is readily available for remobilization once the fluvial energy increases and overcomes the channel's buffering capacity threshold.

Our data show warmer and wetter thaw seasons result in increased stream power during the summer and fall shoulder

**Table 1 Mean ± SD proportions of annual flux from each hydrological period at the CBAWO.**

| | Physically undisturbed headwater slope stream (GS) | | | Physically disturbed headwater slope stream (PT) | | | Main stems (WR and ER) | | |
|---|---|---|---|---|---|---|---|---|---|
| | Nival | Base flow | Pluvial | Nival | Base flow | Pluvial | Nival | Base flow | Pluvial |
| 2005–2017[a] | | | | | | | | | |
| SSC | 74 ± 19% | 19 ± 11% | 8 ± 16% | 67 ± 41% | 2 ± 3% | 14 ± 17% | 84 ± 29% | 1 ± 1% | 24 ± 34% |
| POC | 73 ± 21% | 21 ± 12% | 9 ± 16% | 75 ± 34% | 3 ± 3% | 17 ± 20% | 84 ± 29% | 1 ± 1% | 24 ± 34% |
| DOC | 71 ± 19% | 24 ± 15% | 10 ± 12% | 77 ± 27% | 3 ± 6% | 25 ± 28% | 85 ± 17% | 4 ± 2% | 13 ± 18% |
| TDN | 72 ± 18% | 23 ± 14% | 9 ± 10% | 77 ± 26% | 4 ± 9% | 23 ± 24% | 83 ± 18% | 6 ± 4% | 17 ± 16% |
| Major ions | 69 ± 22% | 28 ± 20% | 7 ± 11% | 67 ± 34% | 9 ± 13% | 28 ± 22% | 76 ± 22% | 8 ± 4% | 28 ± 20% |
| Stream power | 72 ± 17% | 20 ± 17% | 11 ± 5% | 81 ± 17% | 11 ± 8% | 13 ± 10% | 80 ± 17% | 8 ± 8% | 14 ± 12% |
| 2007–2009, 2016 | | | | | | | | | |
| SSC | 37 ± 1% | 40 ± 14% | 20 ± 17% | 53 ± 2% | 18 ± 13% | 30 ± 11% | 37 ± 30% | 4 ± 6% | 59 ± 33% |
| POC | 35 ± 3% | 47 ± 17% | 16 ± 23% | 52 ± 1% | 18 ± 14% | 30 ± 13% | 37 ± 29% | 4 ± 6% | 59 ± 32% |
| DOC | 41 ± 1% | 46 ± 16% | 13 ± 18% | 40 ± 8% | 11 ± 15% | 50 ± 8% | 63 ± 23% | 6 ± 3% | 32 ± 22% |
| TDN | 40 ± 2% | 48 ± 17% | 11 ± 16% | 41 ± 16% | 14 ± 20% | 45 ± 4% | 58 ± 23% | 12 ± 8% | 30 ± 20% |
| Major ions | 42 ± 2% | 45 ± 16% | 12 ± 15% | 37 ± 12% | 24 ± 7% | 40 ± 5% | 41 ± 11% | 12 ± 7% | 47 ± 10% |
| Stream power | 48 ± 12% | 33 ± 12% | 19 ± 1% | 44 ± 14% | 25 ± 3% | 31 ± 11% | 59 ± 13% | 9 ± 4% | 32 ± 10% |

We separate years with significant pluvial responses (2007–2009, 2016) from the remainder of the dataset (2005–2017) to show how shifts in the timing and magnitude of pluvial energy affect proportions of annual flux during each hydrological period. Full flux data are available in the Supplementary Data.
[a]Excluding 2007–2009, 2016.

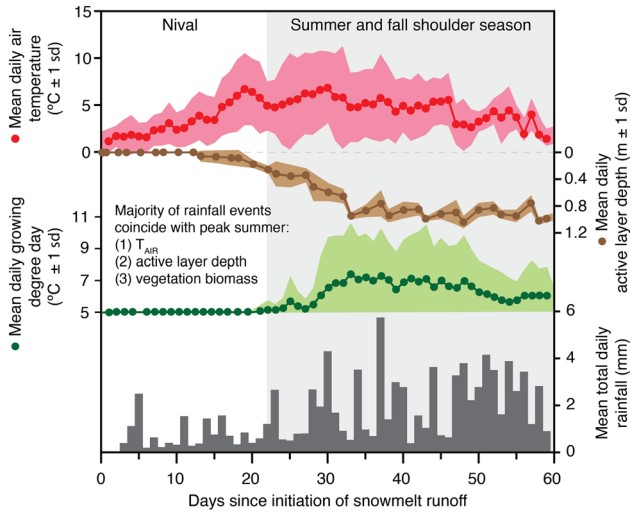

**Fig. 3 Implications of the redistribution of fluvial energy across a lengthening summer thaw season in small High Arctic watersheds (10 km²) underlain by continuous permafrost.** Daily mean (±1 SD) air temperature (2003–2019), active layer depth (2012–2019), growing degree day (GDD; 2003–2019) and total daily rainfall (2003–2019) from the CBAWO. A shift in higher stream powers to later in the summer/fall shoulder season enhances the potential for terrestrial-aquatic connectivity in response to rainfall inputs when annual summer air temperatures, active layer depths, and vegetation productivity are at their annual maximums.

season and (Fig. 1b), in turn, elevated potential for lateral (terrestrial to stream channel) and longitudinal (upstream to downstream) material fluxes along the watershed continuum due to increased terrestrial-aquatic hydrological connectivity. The nival period, when the majority of stream power (76 ± 13% at our study site) is traditionally expended, is characterized by low terrestrial-aquatic hydrological connectivity due to shallow active layer thaw depths (0–5 cm) and variable snow cover (typically 0.1–1.5 m thick) limiting interaction between channelized runoff and terrestrial surfaces (Fig. 3 and Supplementary Fig. 3a). Therefore, the majority of material export during the nival period

is sourced from surface runoff processes (e.g., terrestrial surface materials, temporary channel stores)[28]. Conversely, pluvial runoff couples the terrestrial-aquatic continuum at a time when: (i) the seasonal active layer is deepening (mean JJA active layer depth 0.8 ± 0.3 m), increasing subsurface drainage connectivity[6,30]; and (ii) vegetation productivity is at its peak, increasing above-ground biomass and available organic material from modern terrestrial vegetation[31] (Fig. 3; Supplementary Fig. 3c). Given that the potential for export of terrigenous material is higher during late summer/fall, increased proportions of annual available stream power expended during pluvial events disproportionately increases the erosion and delivery of terrestrial material into the downstream aquatic ecosystem[9,10].

Optical DOM indices measured at the CBAWO support increased levels of terrigenous DOM export during pluvial responses compared to the nival and baseflow periods (2012–2017; Fig. 4; Supplementary Figs. 9 and 10). In headwater catchments, mean $\alpha_{350}$ values were highest during pluvial events (Fig. 4d; Supplementary Fig. 9a); $\alpha_{350}$ values are correlated with lignin phenols (unique biomarkers of vascular plants) and serve as an indicator for terrigenous DOM[32,33]. Concurrent higher growing degree day (GDD; Fig. 3) and DOM freshness index (β:α) values during pluvial responses further support that this DOM was recently-produced in the surrounding terrestrial environment (Fig. 4e; Supplementary Fig. 9c). The lack of vegetative productivity during the nival period, coupled with lower potential for terrestrial interaction and lower DOM β:α values, indicate increased $\alpha_{350}$ values during nival periods compared to baseflow are likely the result of mobilization of terrigenous material from prior growing seasons.

Our results also indicate that the maximum pluvial stream power generated by varying rainfall magnitudes dictates whether terrestrial-aquatic pathways export proportionally more dissolved or particulate material. Low-intensity, short duration rainfall events with resultant stream powers <30 ± 9 W primarily transfer dissolved material (62 ± 25% of the total pluvial flux in only 15 ± 4% of the active flow season; 4–13 days). These data suggest that low- to intermediate hydrological events effectively couple the watershed-scale terrestrial-aquatic continuum for dissolved material and drive the majority of dissolved material export in these streams[34]. In contrast, higher magnitude rainfall

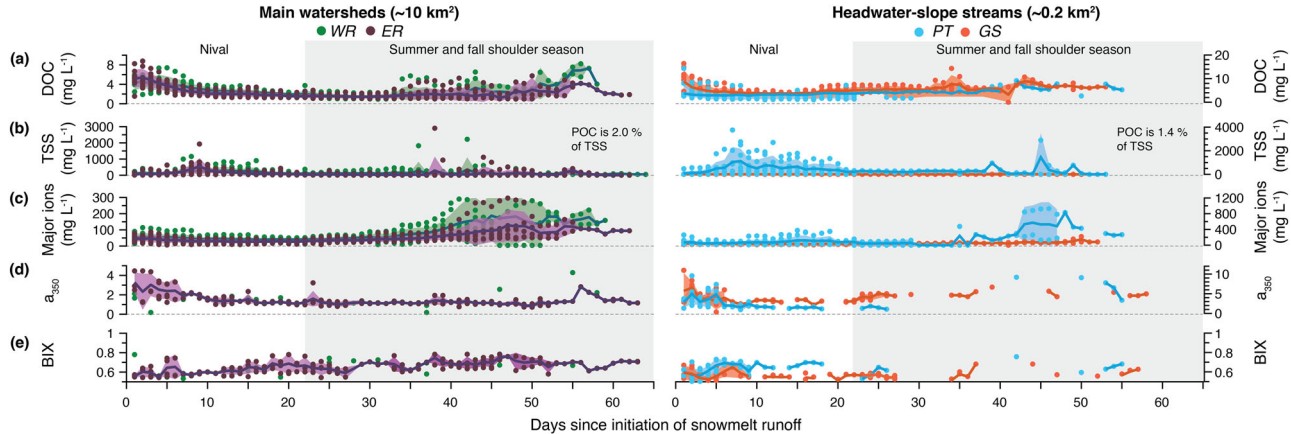

**Fig. 4 Mean daily biogeochemical records normalized to the first day of nival runoff and separated by watershed scale; main watersheds (~10 km²):**
**West River (WR); East River (ER); and headwater-slope streams (~0.2 km²): Ptarmigan (PT); Goose (GS).** (**a**) DOC concentration; (**b**) TSS
concentration (note that POC is 1.4–2.0% of TSS in these systems); (**c**) major ions; (**d**) a350 values, which are an indicator of terrigenous DOM; and (**e**)
BIX values which are an indicator of autochthonous DOM.

events with resultant stream power >30 ± 9 W transfer primarily
particulate material (79 ± 19% of the total pluvial flux in 20 ± 2%
of the active flow season). Spectral slope ratios ($S_R$) of
chromophoric DOM indicated that pluvial runoff from rainfall
events with higher intensities, longer durations, and/or less
frequent recurrence intervals led to higher average molecular
weight DOM in the stream networks[35] (Supplementary Fig. 11).
These patterns suggest that higher pluvial stream powers,
exceeding a threshold of 30 ± 9 W, are necessary to: (i) initiate
channel bank erosion in the upper watershed and (ii) connect
terrestrial-aquatic pathways along the full watershed continuum
for particulate terrigenous materials. It is important to note that
the scale of pluvial responses was strongly controlled by
antecedent conditions. For example, consecutive low-intensity,
longer-duration rainfall events (<0.2 mm h-1, 24–75 h, <2-year
recurrence) can produce higher stream powers >30 W.

Baseflow periods had the lowest stream powers (0.05 ± 0.02 W
in headwaters and 3.0 ± 2.2 W in main streams), with limited
hydrological connectivity and terrestrial to aquatic material
flux[9,10] (Supplementary Fig. 3b). Surface flow in headwater-
slope streams often rapidly ceased following snow exhaustion by
late June or early July, transitioning to a subsurface flow regime
hydrologically coupling the subsurface of terrestrial slopes with
the stream network. Our DOM data show baseflow periods,
despite having the least terrestrial-aquatic connectivity, had the
most in-stream biological activity. In the main streams,
biological/autochthonous index (BIX) values were higher during
the lower-energy baseflow periods, suggesting greater proportions
of recently-produced autochthonous DOM[36]. While BIX values
indicated DOM was primarily of allochthonous origin through-
out our study (range 0.50–0.78), significantly higher BIX values
during baseflow periods (Supplementary Fig. 9b) indicate
proportionally less allochthonous DOM input from the surround-
ing landscape due to limited (sub)surface input and/or more
autochthonous internal DOM cycling. Dissolved nutrient input
from thawing active layer soils during the late season, which is
supported by increased major ion concentrations (Supplementary
Fig. 10d), may further stimulate increased primary productivity
within the stream network[37] at a time when water temperatures
are at their annual maximums (up to 18.2 °C locally). The
enhanced BIX values in the main streams were less evident when
stream powers exceeded ~25 W, suggesting a threshold where
autochthonous DOM production is reduced and/or the auto-
chthonous DOM signal is masked by increased allochthonous

DOM input. As the hydrological season continues to lengthen
and both air and river temperatures increase, in-stream
productivity and increased DOM processing during baseflow
periods will also likely increase, further impacting C processing
potentials in this landscape[38].

Combined, our results show lower magnitude rainfall events
are important mechanisms for supplying Arctic rivers with
dissolved and particulate terrestrial material[9,34]. However, the low
magnitude rainfall events are unable to overcome energy
thresholds needed to transport particulate material downstream
to watershed outlets. This results in the temporary storage of
particulate material within steam channels, where it can be
processed and become available for microbial use during low
energy hydrological periods. When higher magnitude pluvial
events overcame our measured energy thresholds (stream powers
>30 W), particulate material from eroded channel banks and
disturbed hillslopes was directly coupled downstream with the
watershed outlet[10,28]. At this site, rainfall magnitudes great
enough to produce runoff energy to overcome stream power
thresholds only occur on an 8 to 100-year recurrence interval
(<12% chance of exceedance in any given year). The projected
increase in the magnitude and frequency of summer rainfall
events in the Arctic[7,8] suggests these stream power thresholds will
be met with greater frequency, resulting in future increased lateral
particulate export.

**Relative influence of permafrost related geomorphological
disturbance.** In addition to shifts in the timing and magnitude of
fluvial energy, climate warming is causing thaw-induced geo-
morphological disturbance across the Arctic[20,39]. Thaw-induced
geomorphological disturbances are anticipated to occur in over
20% of the permafrost zone, thawing an additional 80 ± 19 Pg C
by 2300 compared to gradual thaw processes alone (e.g., active
layer deepening) and greatly enhancing the permafrost carbon
feedback[20]. These disturbances increase the delivery of terrestrial
materials to aquatic systems[24,25], but the longer-term persistence
(>2–5 years) of these impacts remain relatively unknown. The
paired watersheds in this study were impacted by localized ALDs
in 2007–2008, disturbing 1.2–2.7% of the watersheds' areal extent
(Supplementary Fig. 1). The ALDs (100+) varied from small
hydrologically (dis)connected patches on headwater-slopes and
along channel banks to long (>100 m) linear features that directly
coupled with main watershed streams; prior to 2007 there was no

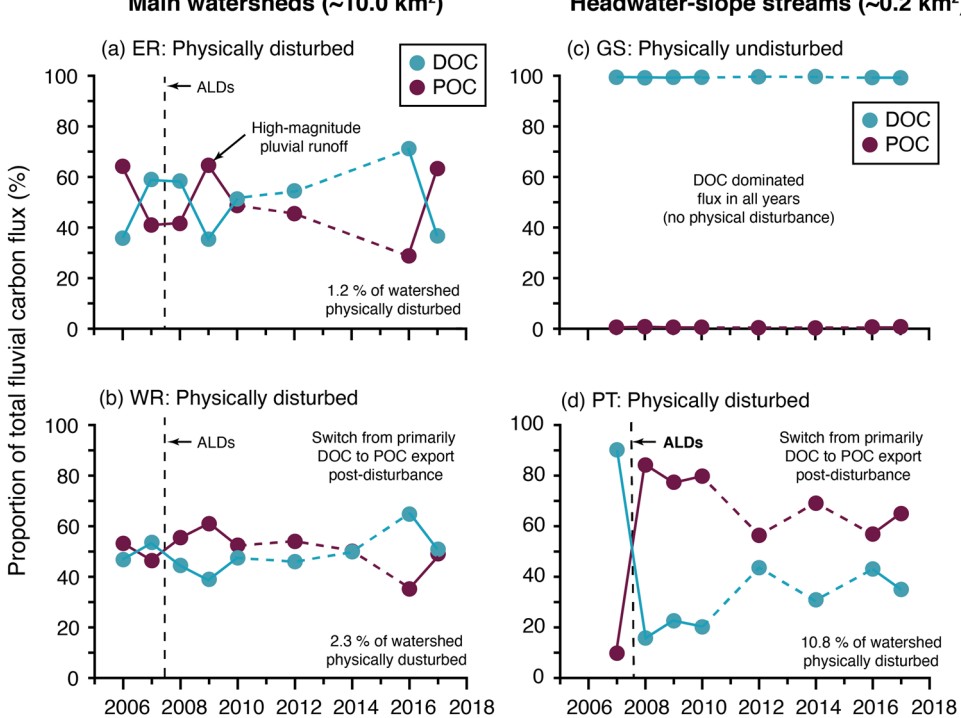

**Fig. 5 The impact of permafrost change on watershed carbon export.** The impact of localized active layer detachments (ALDs; geomorphological disturbance) on the relative contributions (%) of DOC and POC to the total annual fluvial C flux (2006-2017) in: (**a**) East river (ER; 1.2 % of watershed area disturbed); (**b**) West river (WR; 2.7% of watershed area disturbed); (**c**) Goose stream (GS; no geomorphological disturbance); (**d**) Ptarmigan stream (PT; 10.8% of watershed area disturbed). In geomorphologically undisturbed watersheds (**a**), the annual C flux is dominated by DOC; increasing mean annual DOC concentrations from these watersheds were likely the result of watershed greening. Thaw-induced geomorphological disturbance of terrestrial surfaces (a–b,d) decreased annual DOC concentrations at all watershed scales and led to a fundamental shift in the primary form of C export from a DOC- to a POC-dominated flux ($p < 0.05$), with the magnitude and persistence of impact increasing with areal extent of watershed disturbance.

evidence of recent terrestrial disturbance at the CBAWO. Given that warming July-August temperatures are anticipated to increase the initiation of permafrost disturbance in the High Arctic[22], our data provide a unique opportunity to compare the relative influence of changing hydrology versus permafrost thaw-induced geomorphological disturbance on the terrestrial-aquatic continuum in the decade following ALD formation (2007–2017).

We show that despite disturbances covering a relatively small areal extent (<2.7%) of the watershed, processes that reduce DOC availability and transport following permafrost disturbances were strong enough to offset concurrent watershed-scale processes that increased DOC export from vegetated geomorphologically undisturbed headwater-slope streams. DOC concentrations from these undisturbed slopes increased during our study period (Supplementary Fig. 12a; GS). This increase was likely due to combinations of greening (+0.19 to +1.3% yr$^{-1}$ NDVI between 1985–2015[40]) as a result of longer growing seasons leading to increased vegetation biomass[41] and warmer summer temperatures causing higher concentrations of DOM to be released to aquatic ecosystems[42] (Supplementary Figs. 9 and 10). Future changes in Arctic vegetation biomass and productivity with continued climate warming are spatially variable and complex to predict[43–45] and hydrological changes such as changes to the timing of snowmelt may delay vegetation growth in some Arctic systems[46,47]. In contrast to the increased DOC concentrations observed in the undisturbed headwater-slope stream, geomorphological disturbance from ALDs resulted in a decline in interannual DOC concentrations at all watershed scales (Supplementary Fig. 12). This was likely due to a combination of: (i) the geomorphological evolution and stabilization of internal channels within ALDs[9], (ii) enhanced DOC processing within the stream

network[48]; and (iii) preferential sorption of available DOC to newly exposed mineral soils[49,50].

At all watershed scales the ALDs led to a shift in the primary form of C export from a DOC- to a POC-dominated flux, with the magnitude and persistence of impact increasing with the areal extent of watershed disturbance (Fig. 5). Our results show that hydrologically-coupled ALDs need to physically disturb as little as 1.2–2.7% of the watershed area (≤12 km²) to trigger this change in fluvial C export from DOC- to POC-dominated in High Arctic systems. Despite this shift in the form of C export, localized ALDs did not increase the downstream, multiyear watershed-scale flux of particulate material due to fluvial energy limitations causing temporary channel storage[10]. This temporarily stored POC may be made accessible for additional processing during summer baseflow as discussed in the above section.

The lack of multiyear increase in the watershed-scale particulate flux post ALDs was primarily due to insufficient fluvial energy to overcome the thresholds for terrestrial-aquatic connectivity at a time when the additional POC sources are thawed and readily available for mobilization (summer and fall). In this setting, nival runoff is ineffective at mobilizing additional POC exposed by localized ALDs due to limited access to terrestrial material at times of high fluvial energy. To fully couple the POC continuum from disturbed sources to downstream at the catchment outlet, summer rainfall events had to result in stream powers >30 W in the main streams. Pluvial responses with less energy rapidly mobilize and exhaust temporary in-channel particulate stores[28], but are an important mechanism resupplying channel stores with newly mobilized/eroded terrestrial material[9]. This suggests that, at the watershed-scale, the effect of thaw-induced geomorphological disturbance on particulate material

export is smaller than the shift toward a multimodal hydrological regime in this environment. Furthermore, disturbed sites are more vulnerable to rain-induced surface thaw[51], which could further accelerate the additional release of terrestrial materials following future rainfall. Hence, we conclude terrestrial disturbance primes Arctic watersheds for accelerated geomorphological change when future rainfall magnitudes and consequent pluvial responses exceed the current buffering capacity of the terrestrial-aquatic continuum ($30 \pm 9$ W).

In watersheds with a terrestrial disturbance area <1.2%, only high magnitude rainfall events were able to effectively couple particulate flux from headwater channel banks and slopes to the watershed outlet[10] (8–100-year recurrence interval; July 2009; Fig. 5b). It is likely these thresholds differ with increasing watershed area, location of the disturbance within watersheds, hydro-geomorphological connectivity, differing types of geomorphological permafrost disturbance (e.g., retrogressive thaw slumps, thermokarst, thermo-erosion), variability in permafrost conditions (sporadic, discontinuous, continuous), watershed characteristics (e.g., OM content, channel type, vegetation distribution), and continued changes in climate and hydrology. However, as the magnitude and frequency of permafrost thaw-induced geomorphological disturbance intensifies due to climate change, our results suggest hydrologically-coupled disturbances will become an increasingly important mechanism for delivering POC from terrestrial environments into stream networks[52].

Overall, our results show localized ALDs at our site played a stronger role in altering watershed C export along the terrestrial-aquatic continuum than Arctic greening, but a weaker role than increased magnitude and frequency of pluvial events in this energy-limited High Arctic system. In higher-energy Arctic systems, or in instances where permafrost thaw and disturbance are geomorphologically more active (e.g., RTS) or lead to the formation of a talik (former permafrost remaining thawed year-round), geomorphological disturbance itself may play a stronger role in altering C export potentials[20,25,53,54]. However, fluvial energy limitations in the High Arctic indicate channel bank disturbance and hydro-geomorphologically coupled ALDs on terrestrial hillslopes will only result in increased watershed C export if they are coupled with sufficient stream power from pluvial runoff during the summer and fall shoulder season.

**Implications for upscaling to larger watersheds and the circum-Arctic.** Given that watershed attributes in low-order headwater catchments like those at the CBAWO dictate stream chemistry within larger river systems, it is critical to examine changes at the headwater-scale to better integrate and predict changes at larger watershed and pan-Arctic scales[5,55,56]. Our field observations show that the timing and magnitude of available fluvial energy is the key mechanism determining the role stream networks play in transporting and cycling terrigenous C and OM in High Arctic watersheds underlain by continuous permafrost. Our identified energy threshold ($30 \pm 9$ W in the CBAWO main streams), which was necessary to fully connect the terrestrial-aquatic continuum at the watershed-scale, may explain disparities in the literature concerning the role inland waters play in cycling terrestrial C. Arctic aquatic systems where energy thresholds exceed the watersheds' buffering capacity, including the continuum of headwaters to larger river networks, are heavily influenced by allochthonous input and terrestrial connectivity[1]. Lower-energy systems such as small lowland ponds, which represent ~25% of northern surface water area, lack the energy needed to hydrologically connect the terrestrial-aquatic continuum, and therefore primarily internally cycle autochthonous C

and play a minor to negligible role in the mineralization of terrestrial C[57].

In this study, we found evidence of shifts in timing of fluvial energy from a skewed to a multimodal distribution in both large and small circum-arctic watersheds, which aligns with global climate model projections of amplified Arctic warming, decreased winter precipitation, and increased magnitude and frequency of summer rainfall[7,8]. Given that the seasonality of Arctic hydrology is closely linked to regional weather, we can expect the timing and magnitude of this fluvial energy shift to vary among sites. While the spring freshet has been previously recognized as a strong driver of terrestrial-aquatic material export in Arctic inland waters[58,59], we identify the increasing role of rainfall events to enhance terrestrial-aquatic connectivity during periods with the highest annual terrestrial vegetation biomass and deepest active layer depths (Fig. 3). Rain events have also been recently shown to stimulate increased permafrost thaw in Low Arctic systems[51,60], further altering hydrological pathways and terrestrial-aquatic connectivity along the continuum. We further show clear evidence that rainfall events are under-sampled and under-represented in Arctic systems. One option for better capturing rain events is to develop site-specific annual rating curves from integrative parameters measured with in situ optical sensors (e.g., turbidity, major ions, DOM), enabling our understanding of these rainfall events and the late season baseflow we miss each year[5].

Localized ALDs offset the increased DOC export from undisturbed vegetated hillslopes during our study despite covering <2.7% of the watershed areal extent. This indicates changes in C export following permafrost thaw-induced geomorphological disturbance, which occurs in a relatively small area of the watershed, outpaces and offsets changes in C export due to watershed-wide processes such as greening. We identified a threshold where hydrologically-coupled ALDs need to geomorphologically disturb as little as 1.2–2.7% of the watershed area ($\leq 12$ km$^2$) to trigger a fundamental shift in C export regimes from DOC- to POC-dominated. We conclude that ALDs have primed the watershed for accelerated geomorphological change when future rainfall magnitudes and consequent pluvial responses exceed the current buffering capacity of the terrestrial-aquatic continuum in this energy-limited system. It is important to note impacts from ALDs and other localized geomorphologic disturbances might be asynchronous and not additive as the watershed contributing area increases, further highlighting the importance of examining low-order headwater catchments to accurately upscale to larger watershed and pan-Arctic scales.

## Methods

This research was conducted at the Cape Bounty Arctic Watershed Observatory (CBAWO; 2003–2019), located on the south-central coast of Melville Island in the Canadian High Arctic (74°54'N, 109°35'W, Supplementary Fig. 1). This setting is underlain by continuous permafrost (~500 m) and characterized by a polar desert climate (mean annual atmospheric temperature ± 1σ: −14.8 ± 1.3 °C) with limited annual precipitation and runoff (<150 mm yr⁻¹). Mean summer (Jun-Aug; JJA) air temperature was 2.7 ± 2.5 °C (2003–2019). Temperature and rainfall data were obtained from local (2003–2019) and regional (1948–2019) meteorological stations proximal to the CBAWO (Supplementary Fig. 1a). Local rainfall data were used to construct a 72-h intensity-duration-frequency curve for the CBAWO (Gumbel EV1 distribution) to estimate the recurrence interval for each rainfall event (Supplementary Fig. 5).

The CBAWO is comprised of paired watersheds for the West and East (unofficial names; WR and ER, respectively). The West watershed is 8.0 km² and has instrumented headwater slope streams (PT: Ptarmigan and GS: Goose; both ~0.2 km² watershed area), and the East is 11.6 km² (Supplementary Table 1). Both watersheds are non-glacial and have second-order low-sinuosity channels with short reaches of braided, riffle- and step-pool morphology along their lengths. Both rivers have a seasonal, snow-limited flow regime where channel runoff typically begins in early to mid-June and ends in late-August to early-September during freeze-up. Late season rainfall events punctuate baseflow runoff, with pluvial responses largely controlled by the timing and magnitude of rainfall and

antecedent soil moisture conditions. Surface runoff in the upper ER watershed occurs as diffuse flow over vegetated water tracks, hydrologically connecting ponds along the terrestrial-aquatic continuum. Water tracks in the WR watershed are limited to headwater-slope inputs, with the majority of runoff in channelized gravel bed streams.

Instrumentation, datalogger, and analytical method specifications are documented in the Supplementary Methods and Supplementary Table 2. For coherence among records and to provide the finest temporal resolution across all years, we reprocessed all variables from raw data into daily means. To normalize hydrological seasons irrespective of interannual variation in hydrometeorological conditions, we temporally normalized data to the first day of flow instead of calendar dates. All statistical analyses were conducted in Matlab versions *R2016a* and *2020a* software. Correlations were calculated using Pearson's r linear correlation coefficients (Supplementary Tables 3-6). Differences between variables: (i) between watersheds and (ii) between hydrological periods were determined using a combination of ANOVA analyses (Supplementary Table 7) and two-samples $t$ tests (Supplementary Figs. 9–10). Differences in annual stream power distributions were tested using two-sample Kolmogorov–Smirnov tests (Supplementary Table 8). Results from all statistical analyses were considered significant at the $\alpha = 0.05$ level when $p < 0.05$.

Daily JJA active layer depths (Fig. 3) were estimated using ground temperature data collected between 2012–2019 from a 7-m borehole (Supplementary Fig. 1a). Daily active layer depth estimates were calculated using a linear regression of temperatures recorded at 0.3 m and 1.3 m depth, with the depth at which the temperature was 0 °C being the assumed thaw boundary. This method assumed linear decreases in ground temperature within the 1.3 m depth range. Actual active layer depths may vary with local soil moisture and ice content conditions; however, values estimated using linear regression (0.0–1.20 m during JJA) were consistent with mean seasonal maximum active layer depths manually measured using a probe (0.7–1.1 m spatially across both watersheds). Growing degree days (GDD) were calculated as:

$$GDD = T_{max} + T_{min}2 - T_{base} \qquad (1)$$

where $T_{max}$ and $T_{min}$ are the daily maximum and minimum temperatures, respectively, and $T_{base}$ is the base temperature representative of the threshold temperature above which plants are productive ($\geq 5\,°C$).

Stream power represents the rate of energy expenditure along a riverbed and banks:

$$\Omega = \rho g Q s \qquad (2)$$

where stream power ($\Omega$; Watts or kg m$^{-2}$ s$^{-3}$) is the product of water density calculated from measured water temperatures ($\rho$; kg m$^{-3}$), acceleration due to gravity ($g$; 9.8 m s$^{-2}$), discharge (Q; m$^3$ s$^{-1}$), and channel slope ($s$). We applied a 5 m buffer on either side of the middle of the channel in ArcGIS® to represent the channel area, from which the watershed's average slope was calculated (Supplementary Table 1).

## Data availability

Discharge, stream power, DOM optical data, SS, DOC, POC, major ions, and DIN are available in the Supplementary Information. Climate, SS, and Q data for the main watersheds are available as Supplementary Information in Beel et al. (2018). Q, SS, POC, DOC, and major ions from headwater-slope streams are available as Supplementary Information in Beel et al. (2020). This data is also available through the Polar Data Catalog (www.polardata.ca).

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

## Acknowledgements
We thank: past and present members of the CBAWO team for assisting in data collection; S. Liebner, S. Tank, and M. Turetsky for pre-submission feedback on an earlier version of this manuscript. Financial support was provided by ArcticNet, Government of Canada International Polar Year (IPY) and Natural Resources and Engineering Council (NSERC) grants to S.F.L. and M.J.L. Field logistics were provided by the Polar Continental Shelf Program (PCSP), Natural Resources Canada. We thank the Hamlet of Resolute for their permission and support for conducting research at the CBAWO.

## Author contributions
C.R.B. and J.K.H. designed the study, led data analyses and syntheses, and wrote the manuscript. S.F.L., J.F.O., and M.A.P. contributed to study design and data analyses. S.F.L. and M.J.L. led hydrological and biogeochemical data collection. A.J.S. led major ion and nitrogen analyses and synthesis. J.K.Y.H. led the GDD analyses and synthesis. All authors commented on the analysis, interpretation and presentation of the data.

## Competing interests
The authors declare no competing interests.
