## [Peer Review File · Nature Communications]

REVIEWER COMMENTS

Reviewer #1 (Remarks to the Author):

This paper examines long-term variations in hydrologic data from the Cape Bounty Arctic Watershed Observatory in the Canadian High Arctic and considers how changes in the distribution of runoff between spring and summer influences water chemistry. The paper is built around a conceptual model that relates changes in seasonal runoff to terrestrial-aquatic connectivity. The data and ideas presented in the paper are interesting, but I must admit that I found it very difficult to follow. Far too much of the paper is relegated to supplementary material, and what is included in the main text and figures does not stand well on its own. This is perhaps most egregious as related to the water chemistry, where a conceptualized representation of the data are presented in Figure 4, but the reader is forced to visit the supplemental material to see the actual results of the study. I found myself repeatedly having to jump back and forth between the main text/figures and the supplementary material, and in many cases it was hard to see how the actual data translated into the conceptualized representation. I also struggled to understand how decisions were made with respect to categorization of rivers flow regimes into nival-dominated versus multi-modal. The data seem to show a continuum of circumstances rather than two distinct regimes, and the cut-off used to bin data into one category or another is not clearly defined. Furthermore, I struggled to understand how decisions were made about whether or not regime changes had happened over time. The results in the main body of the paper seem to rely very heavily on how patterns in two recent years (2007 and 2009) differ from other years in the long-term record. The supplementary data for other rivers around the Arctic are helpful for arguing that changes in seasonal river discharge are widespread, but as with the main data it is unclear how decisions are being made about whether or not regime change is happening. In general, I think there is too much emphasis placed on the idea that rivers are changing from nival-dominated to bimodal. It seems clear that hydrographs are changing, and that proportional contributions from rainfall are increasing, but whether or not different places have shifted from a nival-dominated to bimodal pattern is less clear and consistent. Lastly, the findings of this paper were, at times, overgeneralized. I'd be cautious about extrapolating to the High Arctic as a whole based on findings from the CBAWO watershed alone, and I'd be even more cautious about extrapolating to the Arctic more generally.

Reviewer #2 (Remarks to the Author):

Review for NCOMMS-20-36810

Article Summary

The manuscript by Beel et al. describes a study using a combination of historic precipitation and discharge records, field-based hydrochemical flux and quality data, and records of active layer disturbance (ALD) to demonstrate that hydrologic connectivity is particularly sensitive to changes in precipitation that occurs in the late thaw season. Additionally, the authors show a shift from an early snowmelt-dominated hydrology to a multimodal distribution, emphasizing a warmer and wetter season will likely result in increased stream power and therefore elevated potential for lateral (terrestrial to stream channel) and longitudinal (upstream to downstream) material fluxes. The authors show this using a "case study" approach with a dataset from the Cape Bounty Arctic Watershed Observatory located in the Canadian Arctic, and indicate that the change in modality can be observed in other high-latitude rivers. Finally, the authors used records of active layer disturbances – which occur in relatively small proportions of the landscape – can fundamentally shift the form of organic matter from dissolved to particulate forms, in a manner that outpaces more distributed and continuous vegetation shifts.

Being familiar with high-latitude systems, such a rigorous assessment of how changes in seasonal precipitation regimes in terms of magnitude, timing, and manner, is necessary to understand the changing responses of Arctic watersheds to climate stressors. While this dataset could be seen as

relatively “short” (2003-2019), it should be considered long-term given the relative difficulty of monitoring high-latitude rivers. In addition, while intuitive, only recently have studies stressed the importance of integrating terrestrial-aquatic interactions (e.g., Tank et al. 2020), as well as hydrology and biogeochemistry (e.g., Vonk et al. 2019), towards constraining high-latitude responses to the direct and indirect impacts of climate change. Therefore, I found that the authors’ integrated assessment of shifting terrestrial-aquatic connectivity, altered hydrology, and resulting biogeochemical signals to be highly novel and would make a valuable contribution to this journal and broader scientific community.

I have several comments and suggestions, mainly focusing on improving clarity of the manuscript for a broader readership beyond the watershed community, that I hope the authors find insightful.

Major Comments

First, I completely understand why an increase in riverine solutes is a cause for concern; however, the broader implication of altered fluxes was not clearly made up-front in the introduction. I suggest that the authors could briefly, but more explicitly, link increasing river solutes to broader global impacts (e.g., Line 46). In other words, the authors should highlight in the first paragraph not only are altered biogeochemical fluxes not measured given the remote nature of these systems (as in Metcalfe et al. 2019) and the lack of long-term records (as in Laudon et al. 2017), that these changes are necessary to parameterize earth system models that predict global change (Kicklighter et al. 2013). While this is mentioned in the last section of the introduction, prioritizing this up-front will emphasize the impacts of altered timing and magnitude of hydrologic connectivity. Adding this detail to the introduction will help emphasize the findings to a broader scientific audience.

Second, the novelty in this paper is that the authors highlight that it matters how precipitation falls – in earlier seasons during the freshet (or nival), there is a likely dominance of surface runoff processes and snowmelt; while in the later seasons, when it rains, there should be increased hydrologic connectivity as a result of the thickened active layer, development of preferential flowpaths, and activation of thermokarst. One major comment that I have for the authors surrounds more clearly defining the idea of an Arctic season, for underscoring a second point: it also matters when this precipitation falls. While it will likely matter when and how precipitation occurs (as the authors discuss) in most if not all high-latitude landscapes, thereby driving hydro-biogeochemical response, there is not a clearly defined seasonality for Arctic systems. That is, the snowmelt period does not happen at the same time everywhere; rather, there is some place-based nuance for when the snowmelt occurs that will require significant effort to meaningfully capture. Therefore, it is important to clarify that these seasons are distinct from our temperate-biased conceptualization of seasons/seasonality. The authors may find the following citations helpful in this case: Olsson et al 2003 Five stages of the Alaskan Arctic cold season with ecosystem implications.

Lastly, the authors include long-term records from other watersheds to investigate changes in modality during warmer or wetter years beyond their primary study catchments, indicating that these processes are fundamentally shifting across high-latitudes. They include data from the Great Rivers Observatory Network as well as other Arctic monitoring sites (e.g., Bonanza Creek, Kuparuk River, De Geerdalen, etc.), which vary widely in contributing area (from <10 km² to >200,000 km²). While data at larger scales is an important indicator of upstream influence, in theory the influence of increased hydrologic connectivity will decrease as contributing area increases, such that the impacts of this bimodal distribution on terrestrial-aquatic connectivity and ALDs will disproportionately influence lateral fluxes in smaller headwater catchments causing longitudinal fluxes to increase (with variability decreasing as water moves further downstream; as in Creed et al. 2015). In this sense, monitoring catchment responses to altered precipitation, season, or disturbance regimes will be most critical at small to intermediate scales (e.g., Shogren et al. 2019), particularly for understanding how materials are pulsed downstream and impact fluxes that are manifested at larger scales (conceptualized in Raymond et al. 2016 and others). It is also important to discuss that as these impacts might

asynchronous and not additive as contributing area increases. Therefore, the conceptualization of hydrologic connectivity from headwaters to watershed outlet continuum should be included in greater detail in the discussion, as while this is intuitive to those that study rivers, it is often less clear to scientists unfamiliar with watershed hydrology.

Minor Comments

Title: I suggest changing the title to: Emerging dominance of summer rainfall driving High Arctic terrestrial-aquatic connectivity

Line 162-164: The idea of low to intermediate-scale hydrologic events driving the majority of stream export seems to align with the idea of an effective discharge (Doyle 2005), which is potentially worth citing here.

Line-By-Line Suggestions

Line 48-50: There have been several studies that have focused primarily on the early freshet, but less focus has been on the later, pre-refreeze season (e.g., Finlay et al. 2006, Holmes et al. 2008, Guo et al. 2007). Perhaps temper the statement "but not quantified" to "but not quantified for most Arctic rivers"?

Line 59: Could cite Laudon et al. 2017 here

Figures & Tables

While S4-S6 are useful in showing the skewed vs. multimodal distributions of discharge, it is somewhat difficult to discern each year's hydrograph from another. I would say that they should remain in the supplement as a resource, but it may be a useful visualization to show the resulting distribution as a colored heat map for each site (on the y axis) by year (x axis) with grey for skewed and the turquoise for multimodal. This would help show the temporal resolution of datasets used for each site (i.e., number of years on record), and also display any historic progression from skewed to multimodal. This would be a figure that could either go in the supplement or in the main text, depending on the authors' preference.

This is a relatively minor comment, but showing the hydrographs (Supplemental Figures S4-S6) using specific discharge (so Q as $m^3/km^2/s$) would enable more direct site to site comparison across sites with varying contributing areas.

Reviewer #3 (Remarks to the Author):

Summary:

This manuscript provides noteworthy and novel findings from a diverse, unique, and temporally extensive dataset from the arctic investigating the relative importance of increased rain dominance versus increased geomorphic disturbance on the future of carbon export via stream systems in these warming environments. The most noteworthy findings of this study are: 1) that although increased geomorphic disturbance – due to warming of permafrost – does impact these environments, the influence of this on carbon export to downstream systems is not consequential unless a threshold level of stream power is met during the summer period, 2) that these needed threshold levels of stream power are now being met more often due to increased summer rainfall observed across the arctic due to climate change, and 3) this is driving a switch away from a single-peak, snowmelt-dominated hydrograph to a bi-modal hydrograph with second peaks also forming during late-summer rain events. The methods of the manuscript appear robust, reproducible, and adequately cited given the standards of my field. The numbers of diverse datasets brought together to support the findings are outstanding, and this makes the overall findings well supported and justified. The manuscript is also extremely well

written, and has obviously been thoroughly edited by several eyes before submission, with no errors in writing that I was able to find. The only issues of concern I see are in the framing of the main points outlined in the abstract, but my concerns here are minor and are outlined in my general comments below. Due to this excellent work in the methods and analysis, the diverse and unique datasets the authors present, and the novel and noteworthy findings of the study, I believe this manuscript merits publication with only minor revisions.

Specific Comments:

In the abstract, if space allows, the authors should include reference to one of their main findings that a threshold ($> 30 W$, in this example) exists for when pluvial streamflow can begin to increase watershed C export in these disturbed High Arctic settings. I see this to be one of the main findings of the study, and something highlighted well by the authors in the section of the discussion focused on this idea of landscapes being primed by increased disturbance, but these disturbances only having an impact on downstream C export if adequate summer rain intensities occur. This point is highly valuable and noteworthy, and I believe should be highlighted more in the abstract.

On Line 24, the clause starting "but play..." needs to be made into a separate sentence. This long sentence needs to be broken up in some way. It is hard to follow as it is currently written with three ideas in one sentence.

– Dr. P. Zion Klos

Please see our responses below to reviewer comments (reproduced in bold) for manuscript NCOMMS-20-36810 "Emerging dominance of summer rainfall in driving High Arctic terrestrial-aquatic connectivity", now titled "Emerging dominance of summer rainfall driving High Arctic terrestrial-aquatic connectivity." We hope this manuscript will now be considered suitable for publication in *Nature Communications*.

Reviewer #1

Comment 1: This paper examines long-term variations in hydrologic data from the Cape Bounty Arctic Watershed Observatory in the Canadian High Arctic and considers how changes in the distribution of runoff between spring and summer influences water chemistry. The paper is built around a conceptual model that relates changes in seasonal runoff to terrestrial-aquatic connectivity. The data and ideas presented in the paper are interesting, but I must admit that I found it very difficult to follow. Far too much of the paper is relegated to supplementary material, and what is included in the main text and figures does not stand well on its own. This is perhaps most egregious as related to the water chemistry, where a conceptualized representation of the data are presented in Figure 4, but the reader is forced to visit the supplemental material to see the actual results of the study. I found myself repeatedly having to jump back and forth between the main text/figures and the supplementary material, and in many cases it was hard to see how the actual data translated into the conceptualized representation.

To address the reviewer's concern regarding too much of our paper being in the supplementary material, we have edited our manuscript so the figures in our main text now show our data, as opposed to conceptual representations of our data. For example, our revised Figure 4 now contains all collected biogeochemical and DOM characterization data for each stream and headwater, as opposed to having the data in the Supplementary Information. Our new Figure 1 has been moved from the Supplementary Information to clearly show how warming local mean summer air temperatures at our site correlate with regional longer-term temperature trends, and in turn: (i) increased normalized difference vegetative index (NVDI) for all vegetation types at the CBAWO; (ii) earlier first day of river flow; and (iii) an increase in the summer thaw season length. We have also moved a table to our main text (now Table 1) to numerically show the redistribution in timings and magnitudes of major hydrological, geomorphological, and biogeochemical fluxes in our watersheds. Further, we have restructured our manuscript to better present and highlight both our key findings and the implications for upscaling to larger watersheds and the circum-Arctic. Please also see our response to Reviewer 3, Comment 1.

Comment 2: I also struggled to understand how decisions were made with respect to categorization of rivers flow regimes into nival-dominated versus multi-modal. The data seem to show a continuum of circumstances rather than two distinct regimes, and the cut-off used to bin data into one category or another is not clearly defined.

We have added a new figure (Supplementary Fig S2) illustrating how we categorized hydrographs into skewed versus multi-modal regimes. This figure shows how annual hydrology was grouped based on subjective observed changes in annual hydrographs. We state in the Methods that differences in annual stream power distributions were tested using two-sample Kolmogorov–Smirnov tests (Lines 685-687). Years classified as nival-dominated (skewed distribution) versus nival- and rainfall-dominated (multi-modal distribution) significantly ($p < 0.05$) differed in all watersheds at the $\alpha = 0.05$ level.

Comment 3: Furthermore, I struggled to understand how decisions were made about whether or not regime changes had happened over time. The results in the main body of the paper seem to rely very heavily on how patterns in two recent years (2007 and 2009) differ from other years in the long-term record. The supplementary data for other rivers around the Arctic are helpful for arguing that changes in seasonal river discharge are widespread, but as with the main data it is unclear how decisions are being made about whether or not regime change is happening.

While the discharge records at the CBAWO are too short to identify longer-term trends in hydrometeorological regimes, we examined longer-term hydrological records from circum-Arctic streams and rivers (including a northern Canadian latitudinal gradient) across both large and small watersheds (8.0 to 2.95 million km² watersheds) in the High and Low Arctic. We have added 4 new figures to the supplementary information (Fig. S2; S4; S6-S8) showing the occurrences of nival-dominated (skewed distribution) versus nival- and rainfall-dominated (multi-modal distribution) annual hydrographs in 12 randomly selected rivers and the 6 largest Arctic rivers during discharge records dating from 1910-2020. Our results show the shift from a nival-dominated to a nival- and pluvial-dominated discharge regime is ubiquitous across the Arctic (and already occurring), although the effects were dampened across larger watersheds and watersheds in the Low Arctic received proportionately more rainfall in general (Lines 146-154). Please also see our response to Reviewer 2, Figs. S5-S6.

Comment 4: In general, I think there is too much emphasis placed on the idea that rivers are changing from nival-dominated to bimodal. It seems clear that hydrographs are changing, and that proportional contributions from rainfall are increasing, but whether or not different places have shifted from a nival-dominated to bimodal pattern is less clear and consistent.

We now clearly show in Supplementary Figs. S6-S8 that the shift from a nival-dominated to a nival- and pluvial-dominated discharge regime we observed at the CBAWO is ubiquitous across the Arctic, although the effects were dampened across larger watersheds and watersheds in the Low Arctic received proportionately more rainfall in general. Please also see our responses to Reviewer 1, Comment 3 and Reviewer 2, Figs. S5-S6. This shift is important because changes in the timing and magnitude of annual discharge patterns and stream power budgets are causing Arctic material transfer regimes to undergo fundamental changes (e.g., Table 1). We use the CBAWO as a case study for changes to the terrestrial-aquatic continuum because: (i) it is the longest integrative data set from the Canadian High Arctic (Lines 143-144) and (ii) it provides a unique opportunity to assess the watershed-scale impacts of localized permafrost thaw-induced geomorphological disturbances on material flux in the decade following their initiation relative to concurrent Arctic greening and shifts in the timing and magnitude of fluvial energy (Lines 269-279). Given that watershed attributes in low-order headwater catchments like those at the CBAWO dictate stream chemistry within larger river systems, it is critical to examine changes at the headwater scale to better integrate and predict changes at larger watershed and pan-Arctic scales (Lines 354-357). Please also see our responses to Reviewer 2, Comments 1 and 5, and Figs. S4-S5.

Comment 5: Lastly, the findings of this paper were, at times, overgeneralized. I'd be cautious about extrapolating to the High Arctic as a whole based on findings from the CBAWO watershed alone, and I'd be even more cautious about extrapolating to the Arctic more generally.

We now discuss how our findings can and cannot be extrapolated in Lines 353-400 of our revised manuscript under the section heading "Implications for upscaling to larger watersheds and the circum-Arctic". Watershed attributes in low-order headwater catchments, like those at the CBAWO, dictate stream chemistry within larger river systems (French et al., 2020). Therefore, it is critical to examine changes at the headwater scale to better integrate and predict changes at larger watershed and pan-Arctic scales. We see evidence of shifts in the timing of fluvial energy from a skewed (snowmelt-dominated) to a multi-modal (snowmelt- and rainfall-dominated) distribution in both large and small circum-arctic watersheds (Figs. S6-S8), aligning with global climate model projections of amplified Arctic warming, decreased winter precipitation, and increased magnitude and frequency of summer rainfall. We acknowledge that the seasonality of Arctic hydrology is closely linked to regional weather, and therefore we can expect the timing and magnitude of this fluvial energy shift to vary among sites (Lines 373-375). We also acknowledge that impacts from ALDs and other localized geomorphologic disturbances might be asynchronous and not additive as the watershed contributing area increases, which further highlights the importance of examining low-order headwater catchments to accurately upscale to larger watershed and pan-Arctic scales (Lines 397-399). Please also see our response to Reviewer 2, Comments 3 and 5.

Reviewer #2

Comment 1: The manuscript by Beel et al. describes a study using a combination of historic precipitation and discharge records, field-based hydrochemical flux and quality data, and records of active layer disturbance (ALD) to demonstrate that hydrologic connectivity is particularly sensitive to changes in precipitation that occurs in the late thaw season. Additionally, the authors show a shift from an early snowmelt-dominated hydrology to a multimodal distribution, emphasizing a warmer and wetter season will likely result in increased stream power and therefore elevated potential for lateral (terrestrial to stream channel) and longitudinal (upstream to downstream) material fluxes. The authors show this using a “case study” approach with a dataset from the Cape Bounty Arctic Watershed Observatory located in the Canadian Arctic, and indicate that the change in modality can be observed in other high-latitude rivers. Finally, the authors used records of active layer disturbances – which occur in relatively small proportions of the landscape – can fundamentally shift the form of organic matter from dissolved to particulate forms, in a manner that outpaces more distributed and continuous vegetation shifts. Being familiar with high-latitude systems, such a rigorous assessment of how changes in seasonal precipitation regimes in terms of magnitude, timing, and manner, is necessary to understand the changing responses of Arctic watersheds to climate stressors. While this dataset could be seen as relatively “short” (2003-2019), it should be considered long-term given the relative difficulty of monitoring high-latitude rivers. In addition, while intuitive, only recently have studies stressed the importance of integrating terrestrial-aquatic interactions (e.g., Tank et al. 2020), as well as hydrology and biogeochemistry (e.g., Vonk et al. 2019), towards constraining high-latitude responses to the direct and indirect impacts of climate change. Therefore, I found that the authors’ integrated assessment of shifting terrestrial-aquatic connectivity, altered hydrology, and resulting biogeochemical signals to be highly novel and would make a valuable contribution to this journal and broader scientific community.

We thank the reviewer for their feedback on our manuscript. In our Introduction, we now highlight that the impacts of climate-related shifts in the timing and magnitude of fluvial energy coupled with terrestrial ecosystem changes are scarcely documented in High Arctic environments due to a lack of integrated longer-term (≥ 10 years) biogeochemical records and the environment’s remote nature (Lines 40-43). Please also see our response to Reviewer 2, Comment 3.

Comment 2: I have several comments and suggestions, mainly focusing on improving clarity of the manuscript for a broader readership beyond the watershed community, that I hope the authors find insightful.

We thank the reviewer for their thoughtful comments and describe how we incorporated their feedback in our responses below.

Comment 3: First, I completely understand why an increase in riverine solutes is a cause for concern; however, the broader implication of altered fluxes was not clearly made up-front in the introduction. I suggest that the authors could briefly, but more explicitly, link increasing river solutes to broader global impacts (e.g., Line 46). In other words, the authors should highlight in the first paragraph not only are altered biogeochemical fluxes not measured given the remote nature of these systems (as in Metcalfe et al. 2019) and the lack of long-term records (as in Laudon et al. 2017), that these changes are necessary to parameterize earth system models that predict global change (Kicklighter et al. 2013). While this is mentioned in the last section of the introduction, prioritizing this up-front will emphasize the impacts of altered timing and magnitude of hydrologic connectivity. Adding this detail to the introduction will help emphasize the findings to a broader scientific audience.

We now state in the first paragraph of our Introduction (Lines 33-48) that accurate biogeochemical budgets across full hydrological seasons are critical reference points for earth system models and are required to predict the strength and timing of climate feedback mechanisms in the Arctic, including the permafrost carbon feedback. Therefore, quantifying the impacts of altered timing and magnitude of hydrological connectivity in these environments are necessary to parameterize earth system models that predict global change. However, we note, the impacts of climate-related shifts in the timing and magnitude of fluvial energy coupled with terrestrial ecosystem changes are scarcely documented in these environments due to a lack of integrated longer-term (≥ 10 years) biogeochemical records and the remote nature of the High Arctic.

Comment 4: Second, the novelty in this paper is that the authors highlight that it matters how precipitation falls – in earlier seasons during the freshet (or nival), there is a likely dominance of surface runoff processes and snowmelt; while in the later seasons, when it rains, there should be increased hydrologic connectivity as a result of the thickened active layer, development of preferential flowpaths, and activation of thermokarst. One major comment that I have for the authors surrounds more clearly defining the idea of an Arctic season, for underscoring a second point: it also matters when this precipitation falls. While it will likely matter when and how precipitation occurs (as the authors discuss) in most if not all high-latitude landscapes, thereby

driving hydro-biogeochemical response, there is not a clearly defined seasonality for Arctic systems. That is, the snowmelt period does not happen at the same time everywhere; rather, there is some place-based nuance for when the snowmelt occurs that will require significant effort to meaningfully capture. Therefore, it is important to clarify that these seasons are distinct from our temperate-biased conceptualization of seasons/seasonality. The authors may find the following citations helpful in this case: Olsson et al 2003 Five stages of the Alaskan Arctic cold season with ecosystem implications.

The reviewer raises an important point that Arctic seasonality differs from temperate-based seasonality. We now define Arctic seasonality and hydrological regimes in Lines 87-101 of the revised manuscript, prior to discussing our finding that Arctic systems are already transitioning from a primarily skewed (nival-) to a multi-modal (nival- and pluvial-dominated) hydrological regime. We reference Olsson et al. (2003) to emphasize that the primary control on hydrology in Arctic watersheds is seasonal phase changes in water, characterized by a short summer flow season between spring break-up and fall freeze-up (Lines 87-90). We also now explicitly state that unique seasonality of Arctic hydrology varies with latitude and is closely linked to regional weather (Lines 103-104), and that we can expect the timing and magnitude of this fluvial energy shift to vary among sites (Lines 373-375).

To emphasize that it matters when and how runoff occurs (from snowmelt versus rainfall runoff), we now explicitly state increased stream power during the summer and fall shoulder season elevates potential for lateral (terrestrial to stream channel) and longitudinal (upstream to downstream) material fluxes along the watershed continuum due to increased terrestrial-aquatic hydrological connectivity (Lines 172-175). We describe that the nival period is characterized by low terrestrial-aquatic hydrological connectivity due to shallow active layer thaw depths and variable snow cover limiting interaction between channelized runoff and terrestrial surfaces; thus, the majority of material export during the nival period is sourced from surface runoff processes (e.g., shallow-thawed terrestrial sources, temporary channel stores; Lines 176-182). Conversely, we now state, pluvial runoff couples the terrestrial-aquatic continuum when the seasonal active layer is deepening, increasing subsurface drainage connectivity, and vegetation productivity is at its peak, increasing above-ground biomass and available organic material from modern terrestrial vegetation (Lines 182-189).

Comment 5: Lastly, the authors include long-term records from other watersheds to investigate changes in modality during warmer or wetter years beyond their primary study catchments, indicating that these processes are fundamentally shifting across high-latitudes. They include data from the Great Rivers Observatory Network as well

as other Arctic monitoring sites (e.g., Bonanza Creek, Kuparuk River, De Geerdalen, etc.), which vary widely in contributing area (from <10 km² to >200,000 km²). While data at larger scales is an important indicator of upstream influence, in theory the influence of increased hydrologic connectivity will decrease as contributing area increases, such that the impacts of this bimodal distribution on terrestrial-aquatic connectivity and ALDs will disproportionately influence lateral fluxes in smaller headwater catchments causing longitudinal fluxes to increase (with variability decreasing as water moves further downstream; as in Creed et al. 2015). In this sense, monitoring catchment responses to altered precipitation, season, or disturbance regimes will be most critical at small to intermediate scales (e.g., Shogren et al. 2019), particularly for understanding how materials are pulsed downstream and impact fluxes that are manifested at larger scales (conceptualized in Raymond et al. 2016 and others). It is also important to discuss that as these impacts might asynchronous and not additive as contributing area increases. Therefore, the conceptualization of hydrologic connectivity from headwaters to watershed outlet continuum should be included in greater detail in the discussion, as while this is intuitive to those that study rivers, it is often less clear to scientists unfamiliar with watershed hydrology.

We now define and describe in detail the terrestrial-to-aquatic watershed continuum and its controls in Lines 156-170. In the revised manuscript, we have included a new section “Implications for upscaling to larger watersheds and the circum-Arctic” (Lines 353-400). We introduce this section by stating how it is critical to examine changes at the headwater scale to better integrate and predict changes at larger watershed and pan-Arctic scales, given that watershed attributes in low-order headwater catchments like those at the CBAWO dictate stream chemistry within larger river systems (Lines 354-357). We note that impacts from both changing hydrological regimes and localized geomorphologic disturbances might be asynchronous and not additive as the watershed contributing area increases, further highlighting the importance of examining low-order headwater catchments to accurately upscale to larger watershed and pan-Arctic scales (Lines 397-400).

Title: I suggest changing the title to: Emerging dominance of summer rainfall driving High Arctic terrestrial-aquatic connectivity

We have changed the title to “Emerging dominance of summer rainfall driving High Arctic terrestrial-aquatic connectivity”, as per the reviewer’s suggestion.

Line 162-164: The idea of low to intermediate-scale hydrologic events driving the majority of stream export seems to align with the idea of an effective discharge (Doyle 2005), which is potentially worth citing here.

We now reference Doyle (2005) in Lines 211 and 249 of the revised manuscript.

Line 48-50: There have been several studies that have focused primarily on the early freshet, but less focus has been on the later, pre-refreeze season (e.g., Finlay et al. 2006, Holmes et al. 2008, Guo et al. 2007). Perhaps temper the statement “but not quantified” to “but not quantified for most Arctic rivers”?

This change has been made as suggested (Line 52).

Line 59: Could cite Laudon et al. 2017 here

We now reference Laudon et al. (2017) in Lines 42 and 61 of the revised manuscript.

Figs. S5-S6: While S4-S6 are useful in showing the skewed vs. multimodal distributions of discharge, it is somewhat difficult to discern each year’s hydrograph from another. I would say that they should remain in the supplement as a resource, but it may be a useful visualization to show the resulting distribution as a colored heat map for each site (on the y axis) by year (x axis) with grey for skewed and the turquoise for multimodal. This would help show the temporal resolution of datasets used for each site (i.e., number of years on record), and also display any historic progression from skewed to multimodal. This would be a figure that could either go in the supplement or in the main text, depending on the authors’ preference.

As per the reviewer’s suggestion, we have reformatted these figures (now Figs. S6-S8) to show the occurrence of skewed versus multimodal distributions for the complete historical discharge records of each river selected. Please also see our response to Reviewer 1, Comment 3.

Figs. S4-S6: This is a relatively minor comment, but showing the hydrographs (Supplemental Figures S4-S6) using specific discharge (so Q as $m^3/km^2/s$) would enable more direct site to site comparison across sites with varying contributing areas.

These figures now show heatmaps of skewed versus multimodal distributions for the complete historical discharge records of each river, as opposed to individual hydrographs for selected years. Please also see our response to Reviewer 2, Figs. S5-S6.

Reviewer #3

Comment 1: This manuscript provides noteworthy and novel findings from a diverse, unique, and temporally extensive dataset from the arctic investigating the relative importance of increased rain dominance versus increased geomorphic disturbance on the future of carbon export via stream systems in these warming environments. The most noteworthy findings of this study are: 1) that although increased geomorphic disturbance – due to warming of permafrost – does impact these environments, the influence of this on carbon export to downstream systems is not consequential unless a threshold level of stream power is met during the summer period, 2) that these needed threshold levels of stream power are now being met more often due to increased summer rainfall observed across the arctic due to climate change, and 3) this is driving a switch away from a single-peak, snowmelt-dominated hydrograph to a bi-modal hydrograph with second peaks also forming during late-summer rain events. The methods of the manuscript appear robust, reproducible, and adequately cited given the standards of my field. The numbers of diverse datasets brought together to support the findings are outstanding, and this makes the overall findings well supported and justified. The manuscript is also extremely well written, and has obviously been thoroughly edited by several eyes before submission, with no errors in writing that I was able to find. The only issues of concern I see are in the framing of the main points outlined in the abstract, but my concerns here are minor and are outlined in my general comments below. Due to this excellent work in the methods and analysis, the diverse and unique datasets the authors present, and the novel and noteworthy findings of the study, I believe this manuscript merits publication with only minor revisions.

We thank the reviewer for their feedback. To better highlight our key findings, we have restructured our results and discussion into four new sections: (1) “Shifting magnitude and distribution of fluvial energy”; (2) “Timing and magnitude of fluvial energy controls terrestrial-aquatic connectivity”; (3) “Relative influence of permafrost related geomorphological disturbance”; and (4) “Implications for upscaling to larger watersheds and the circum-Arctic.” We have also re-written our abstract (Lines 18-31) to emphasize these findings. Please also see our response to Reviewer 3, Comment 2.

Comment 2: In the abstract, if space allows, the authors should include reference to one of their main findings that a threshold (> 30 W, in this example) exists for when pluvial streamflow can begin to increase watershed C export in these disturbed High Arctic settings. I see this to be one of the main findings of the study, and something highlighted well by the authors in the section of the discussion focused on this idea of landscapes being primed by increased disturbance, but these disturbances only having an impact on downstream C export if adequate summer rain intensities occur.

This point is highly valuable and noteworthy, and I believe should be highlighted more in the abstract.

We now state in the abstract: “To overcome the watersheds’ buffering capacity for particulate material (30 ± 9 W), rainfall events had to increase by an order of magnitude...” (Lines 27-29).

On Line 24, the clause starting “but play...” needs to be made into a separate sentence. This long sentence needs to be broken up in some way. It is hard to follow as it is currently written with three ideas in one sentence.

We have now edited these ideas to read: “Increased late summer rainfall enhanced terrestrial-aquatic connectivity for dissolved (e.g., DOC) and particulate (e.g., POC) material fluxes. Permafrost disturbances (< 3 % of the watersheds’ areal extent) reduced watershed-scale DOC export, offsetting concurrent increased DOC export in undisturbed watersheds. To overcome the watersheds’ buffering capacity for particulate material (~30 W), rainfall events had to increase by an order of magnitude, indicating the landscape is primed for accelerated geomorphic change when future rainfall magnitudes and consequent pluvial responses exceed the current buffering capacity of the terrestrial-aquatic continuum.” (Lines 24-31)

REVIEWERS' COMMENTS

Reviewer #1 (Remarks to the Author):

Nice job revising this manuscript. My primary concerns have been addressed, and I think this a valuable contribution. It is still a bit difficult to fully embrace and appreciate the work with so much important information relegated to the supplementary section, but I am now comfortable with the main points emphasized in the manuscript (and the analytical approach used to underpin those points). I don't have any additional suggestions for revisions at this juncture.

Reviewer #2 (Remarks to the Author):

Review of NCOMMS-20-36810A – Remarks to the Authors

Summary

The authors by Beel, Heslop et al. describe a detailed study of the Cape Bounty Arctic Watershed Observatory (CBAWO). The authors find that in their study watersheds, with records from 2003-2017, there was a transition from a snowmelt-dominated hydrologic regime to a rain-fall dominated (or multi-model) hydrologic regime. The authors discuss that the transition of when water has the most power fundamentally shifts the mobilization patterns of dissolved and particulate carbon (DOC and POC), in addition to other biotic and geogenic solutes.

Upon this revision, I do want to commend the authors on a thoughtful and thorough response to my comments and on a greatly improved manuscript. The authors did a nice job of responding to my concerns, and have added sections of text, included further context (e.g., citations), and reworked the figures (both aesthetically and in terms of organization), and the addition of section headings in the discussion was helpful in honing the authors main messages. I do have several minor comments for the authors to consider, which are meant to improve the manuscript's clarity. Otherwise, I look forward to seeing the manuscript in print, as the authors' work is both timely and compelling.

Minor Comments

If possible, I recommend defining the units of W at Line 28.

Line 76: Please put "indices that indicate DOM origin" before the parentheses, to improve the clarity of this sentence.

Line 87: Perhaps instead of saying "water", refining this term to "hydrology", as water form and movement is inferred.

Line 177: Add "and" to "is characterized"

Line 182: I recommend adding a citation to this effect... Could potentially cite the conceptualization by Neilson et al. 2018 here.

Line 204 (and throughout): While I don't want to get bogged down in semantics, it might be a bit more accurate to temper the results to "indicate" vs. "show", particularly when results are taken from integrated measurements (in this case, lateral transport / connectivity is inferred and not directly measured). This isn't the case for the section starting at Lines 281, when ALDs are directly measured.

Line 272: What is meant by (dis)connected? It might be more precise to say "heterogeneously connected patches".

Line 364: Suggest rephrase to avoid parentheticals: "Arctic aquatic systems where energy thresholds

exceed the watersheds' buffering capacity, including the continuum of headwaters to larger river networks, are heavily influenced by allochthonous..."

Line 365: As the comment above, suggest rephrase to: "Lower energy systems such as small ponds, which represent ~25% of northern surface water area, lack the necessary energy to hydrologically connect..."

Line 370: Rephrase to: "In this study, we found evidence of shifts in timing from a skewed to a multimodal distribution in both small and large circum-Arctic watersheds..."

Line 381-382: This sentence is important! I suggest a slight rewrite to "Further, we show clear evidence that rainfall events are under-sampled and under-represented in Arctic systems"

Figures & Tables:

I really appreciated the move of Figure 1 from the supplement. It presents clearer context for the study.

I am curious if Figure 4 could be split into a two-panel figure, with main watersheds and headwater slope streams presented side by side. Because several of the axes are different scales, it is currently challenging to discern differences between watershed DOC, TSS, etc between these groupings. My other suggestion for the authors to make sure that this figure is to modify and make it color-blind friendly (e.g., if the authors wish to keep the color scheme, simply changing the symbols would ensure that all will be able to read the figure adequately).

REVIEWERS' COMMENTS

Reviewer #1 (Remarks to the Author):

Nice job revising this manuscript. My primary concerns have been addressed, and I think this a valuable contribution. It is still a bit difficult to fully embrace and appreciate the work with so much important information relegated to the supplementary section, but I am now comfortable with the main points emphasized in the manuscript (and the analytical approach used to underpin those points). I don't have any additional suggestions for revisions at this juncture.

We thank the reviewer for their time and comments on our manuscript, their input has greatly improved this paper.

Reviewer #2 (Remarks to the Author):

Review of NCOMMS-20-36810A – Remarks to the Authors

Summary

The authors by Beel, Heslop et al. describe a detailed study of the Cape Bounty Arctic Watershed Observatory (CBAWO). The authors find that in their study watersheds, with records from 2003-2017, there was a transition from a snowmelt-dominated hydrologic regime to a rain-fall dominated (or multi-model) hydrologic regime. The authors discuss that the transition of when water has the most power fundamentally shifts the mobilization patterns of dissolved and particulate carbon (DOC and POC), in addition to other biotic and geogenic solutes.

Upon this revision, I do want to commend the authors on a thoughtful and thorough response to my comments and on a greatly improved manuscript. The authors did a nice job of responding to my concerns, and have added sections of text, included further context (e.g., citations), and reworked the figures (both aesthetically and in terms of organization), and the addition of section headings in the discussion was helpful in honing the authors main messages. I do have several minor comments for the authors to consider, which are meant to improve the manuscript's clarity. Otherwise, I look forward to seeing the manuscript in print, as the authors' work is both timely and compelling.

We thank the reviewer for their time and comments on our manuscript, and describe how we have addressed their remaining comments below.

Minor Comments

If possible, I recommend defining the units of W at Line 28.

We now define W as Watt within the text (Line 29).

Line 76: Please put "indices that indicate DOM origin" before the parentheses, to improve the clarity of this sentence.

We have altered the sentence as per the reviewer's suggestion (Lines 76-77).

Line 87: Perhaps instead of saying "water", refining this term to "hydrology", as water form and movement is inferred.

We have amended this sentence to read “The primary control on Arctic hydrology is seasonal phase changes in water form and movement...” (Lines 89-90)

Line 177: Add “and” to “is characterized”

We have opted to keep our original phrasing, as “The nival period... is characterized...” is grammatically correct and makes the sentence clearer (Lines 178-179).

Line 182: I recommend adding a citation to this effect... Could potentially cite the conceptualization by Neilson et al. 2018 here.

We now cite Favaro and Lamoureux (28 in our reference list) here, whose work demonstrated these processes (Line 184).

Line 204 (and throughout): While I don’t want to get bogged down in semantics, it might be a bit more accurate to temper the results to “indicate” vs. “show”, particularly when results are taken from integrated measurements (in this case, lateral transport / connectivity is inferred and not directly measured). This isn’t the case for the section starting at Lines 281, when ALDs are directly measured.

We have changed “show” to “indicate” (Line 206).

Line 272: What is meant by (dis)connected? It might be more precise to say “heterogeneously connected patches”.

The term “(dis)connected” is common in geomorphology and hydrology literature to simultaneously refer to both “dis-connected” systems (e.g., no hydrological coupling) and “connected” systems which are hydrologically coupled. We have chosen to keep this term in the manuscript, as it is more accurate and concise (Line 276).

Line 364: Suggest rephrase to avoid parentheticals: “Arctic aquatic systems where energy thresholds exceed the watersheds’ buffering capacity, including the continuum of headwaters to larger river networks, are heavily influenced by allochthonous...”

We have amended this sentence as per the reviewer’s suggestion (Lines 367-369).

Line 365: As the comment above, suggest rephrase to: “Lower energy systems such as small ponds, which represent ~25% of northern surface water area, lack the necessary energy to hydrologically connect...”

We have changed this sentence as suggested by the reviewer (Lines 369-373).

Line 370: Rephrase to: “In this study, we found evidence of shifts in timing from a skewed to a multimodal distribution in both small and large circum-Arctic watersheds...”

We have edited this sentence as per the reviewer’s suggestion (Lines 375-378).

Line 381-382: This sentence is important! I suggest a slight rewrite to “Further, we show clear evidence that rainfall events are under-sampled and under-represented in Arctic systems”

We have amended the sentence as per the reviewer's suggestion (Lines 395-397).

Figures & Tables:

I really appreciated the move of Figure 1 from the supplement. It presents clearer context for the study.

I am curious if Figure 4 could be split into a two-panel figure, with main watersheds and headwater slope streams presented side by side. Because several of the axes are different scales, it is currently challenging to discern differences between watershed DOC, TSS, etc between these groupings. My other suggestion for the authors to make sure that this figure is to modify and make it color-blind friendly (e.g., if the authors wish to keep the color scheme, simply changing the symbols would ensure that all will be able to read the figure adequately).

We thank the reviewer for their comments. We have amended the shaded areas in the main manuscript file to make the figures more visually accessible. To increase the clarity of Figure 4, we have split the figure into two panels (main watersheds and headwater slopes) as per the reviewer's suggestion.